# Generation of Glioblastoma Patient-Derived Intracranial Xenografts for Preclinical Studies

**DOI:** 10.3390/ijms21145113

**Published:** 2020-07-20

**Authors:** Amber E. Kerstetter-Fogle, Peggy L. R. Harris, Susann M. Brady-Kalnay, Andrew E. Sloan

**Affiliations:** 1Department of Neurological Surgery, Case Western Reserve University and University Hospitals Cleveland, OH 44106, USA; plr2@case.edu (P.L.R.H.); andrew.sloan@uhhospitals.org (A.E.S.); 2Case Comprehensive Cancer Center, Case Western Reserve University, Cleveland, OH 44106, USA; smb4@case.edu; 3Department of Molecular Biology & Microbiology, School of Medicine, Case Western Reserve University, Cleveland, OH 44106, USA; 4Seidman Cancer Center University Hospitals, Cleveland, OH 44106, USA

**Keywords:** GBM, intracranial, patient-derived xenograft, preclinical

## Abstract

Glioblastoma multiforme (GBM) is the most malignant primary brain cancer affecting adults. Therapeutic options for GBM have remained the same for over a decade with no significant improvement. Many therapies that are successful in culture have failed in patients, likely due to the complex microenvironment in the brain, which has yet to be reproduced in any culture model. Furthermore, the high passage number of cultured cells and clonal selection fail to recapitulate the molecular and genomic signatures of GBM. We have established orthotopic patient-derived xenografts (PDX) from 37 GBM patients with human GBM. Of the 69 patient samples analyzed, we were successful in passaging 37 lines three or more generations (53.6%). After phenotypic characterization of the xenografted tumor tissue, two different growth patterns emerged highly invasive or localized. The phenotype was dependent on malignancy and previous treatment of the patient from which the xenograft was derived. Physiologically, mice exhibited symptoms more quickly with each subsequent passage, particularly in the localized tumors. Study of these physiologically relevant human xenografts in mice will enable therapeutic screenings in a microenvironment that more closely resembles GBM and may allow development of individualized patient models which may eventually be used for simulating treatment.

## 1. Introduction

Glioblastoma multiforme (GBM) is the most common malignant primary brain tumor in adults, comprising 54% of all gliomas [1]. Despite recent advances, the median survival remains 14–16 months with current treatment paradigms that include surgically debulking the tumor, chemotherapy, and radiation. Successful treatments are inhibited by the very nature of the disease, invasiveness prevents complete surgical removal and promotes resistance to chemotherapy and radiation which limit the effectiveness of the available treatments [2,3,4].

A link between brain tumors and fetal development has long been recognized; however, recently several groups have discovered that brain tumor cells display features similar to those of fetal stem cells upon transplantation [5,6,7]. Glioma stem cells (GSCs) are defined by their ability to self-renew and propagate tumors in vivo, with the capacity for angiogenesis and invasion [8]. GSCs have been extensively characterized, and appear to be resistant to the conventional therapy treatments of radiation and chemotherapy. These cells appear to remain following treatment and contribute to the recurrence of the disease [2,3,9].

Molecularly-targeted therapies for the treatment of GBM have largely failed, in part due to the extreme heterogeneity of the disease and the numerous cell types that contribute to the tumor. For instance, therapies directed towards epidermal growth factor receptor (EGFR), platelet derived growth factor receptor (PDGFR), tyrosine protein kinase (MET), phosphatase and tensin homolog (PTEN) have failed in part due to this heterogeneity. With this in mind, multiple lines of investigation must be pursued to develop therapy strategies, vaccine therapy, and/or models of GBM, to help facilitate the investigation of detailed molecular and cellular events associated with tumor progression.

Each experimental model contains advantages and disadvantages. For instance, in vitro 2-dimensional models of GBM are feasible and can be established easily and quickly from primary tumors allowing for numerous evaluations, including genetic analysis and drug screenings. However, clonal selection in vitro and the lack of a tumor microenvironment make it difficult to tease out the mechanisms underlying tumor progression [10]. Maintenance of the common hallmarks of the primary tumor, such as mutations and amplifications, are usually lost in the in vitro culture over time with increased passage due to genetic instability and inadvertent selection of more aggressive subpopulations of cells [11,12,13,14]. Heterogeneity and genomic alterations however, can still be maintained in orthotopic xenograft models of GBM. In vivo heterotopic models of GBM are typically passaged through the flank/subcutaneous model and do not take into effect the unique microenvironment of the brain [15,16,17]. Studying cancer biology in vitro and outside of the brain environment, limits the potential therapeutic value of these models. The high failure rate of currently-tested therapies in patients may be due to the screening processes presently utilized in such in vitro and heterotopic models.

Preclinical models of GBM historically utilized immortalized cell lines (U87, T-98, etc.) injected intracranially or fresh cells or tissue passaged using a flank/subcutaneous model. These heterotopic models are easy to work with and readily allow simple measurement of subcutaneous tumor, but grow as localized masses similar to brain metastasis rather than invasive masses like human GBM. Thus, the failure of the preclinical models is believed to be due in part to the vast differences between human GBM and tumor models based on immortalized cell lines or heterotopic models which do not mimic the brain microenvironment.

In contrast, however, GSCs propagated as orthotopic tumors in immunocompromised hosts continue to maintain histological and genetic similarity with the human GBM from which they arise [6]. Establishment of patient-derived orthotopic intracranial xenograft (PDX) models for screening potential treatments for GBM could increase the success rate for preclinical studies. Here we successfully developed intracranial PDX models that maintain the properties and grow in a microenvironment similar to that of the original tumor.

## 2. Results

Sixty-nine GBM samples were collected for PDX generation from our institution between 2010 and 2016. Patient ages ranged from 34 to 84 years with an average age of 60.8 years, with male patients predominating in our sample population as is typical according to the Cerebral Brain Tumor Registry of the United States (CBTRUS). After MRI imaging indicating a potential GBM, the tumor was resected and the GBM tissue was processed to derive tumor stem-like cells (Figure 1). Our study spanned several years and included tumors from multiple spatial locations within the brain as well as both newly-diagnosed and recurrent tumors. Tumors generating neurospheres in culture had a higher take rate than single cell cultures. Samples reflected the standard of care of radiation and chemotherapy, as well as primary and recurrent tumors, all factors which may contribute to the successful implantation rate of the samples into the mouse brain. We selected samples that established neurospheres by day three of the in vitro culture. Occasionally, we would observe primary GBM cells adhering to the suspension plates (Figure 2A,C,D). The take rate of the culture was not dependent on the morphology of the cells at early culture. 

Case 2025 (Figure 2A) was a confirmed GBM and resulted in adherent neurospheres with small differentiated-like cells adhering to the suspension culture flask. Occasionally, as was the case with sample 2033, we would obtain large 3-dimensional neurospheres which would also result in tumor growth in an NOD scid gamma (NSG) mouse (Figure 2B). More often, early cultures from recurrent (Figure 2C) or newly diagnosed cases (Figure 2D) resulted in clumps of neurosphere-like cells with a few adherent cells stuck to the suspension flask. All in vitro growth patterns resulted in orthotopic xenografts.

Of the original 69 patient samples injected intracranially into mice, 37 tumors contained viable cells after the 1st passage allowing for additional passaging into mice. The 37 tumor samples were consecutively passaged a minimum of 3 times into mice. Of the 37 tumors studied, 11 (15.9%), had a 40% or more increase in tumor growth rate between the 1st and 3rd intracranial passage in animals (Table 1).

In 18 of the 37 cases (48.9%), mice bearing xenografts succumbed to the tumor 9–38% more quickly after the third passage of the tumor than the first, but in 8 (21.6%) the tumor did not appear to become more malignant after serial passage (Table 1). Age at diagnosis and gender did not seem to play a role in the take rate of the cells. To note, most samples were of Isocitrate dehydrogenase-1 (IDH1) wild type molecular status and all were grade IV glioma. Our lab, along with others have reported difficulty propagating IDH1R132H mutant cultures [18]. Mouse antigen cells were removed from the culture with the mouse cell depletion kit from Miltenyi Biotec, and human tumor stem-like cells were engrafted again into at least two NSG mice indicated by the “ic” passage number (ic1, ic2, or ic3). The average time needed for the first passage was 125 days, 100 days for the second passage, and 89 for the third passage through an immunosuppressed mouse. Overall, the growth of cells increased i.e., towards more malignant, as the in vivo passage number increased. Cells taken directly from the mouse were also able to form neurospheres in culture quicker as the passage number increased from first to second and then to third.

### Primary GBM and PDX Derivatives Morphology After In Vivo Passaging

Tissue sections from successful PDX models were stained with IDH1R132H, Ki67 and ATRX to confirm molecular similarities with parental tumor (Figure 3). Tissue sections models were also subsequently stained with hematoxylin and eosin (H&E,) and immunostained for human cytoplasmic marker (STEM 121) and human nuclear antigen to identify and assess the number and morphology of the human cells which were engrafted.

Control mice not receiving human tumor cells showed no positive cells for human-specific antigens, IDH1, Ki67 or ATRX (data not shown). A couple of representative cases are described. Case 2409 was derived from a 57-year-old patient undergoing resection after a confirmed new GBM diagnosis (see Figure 1). Interestingly, case 2409 lacked the IDH1R132H mutations in the parental tumor, which was recapitulated in the PDX model, consistent with the possible characterization of being IDH1wt which was determined in the clinic (Figure 3A,E). Further, the PDX model exhibited increased Ki67 positive cells within the tumor consistent with the parental tumor (Figure 3B,F). ATRX immunohistochemistry was consistent in the mouse and the human PDX across the 15 samples tested. In the representative sample of 2409, ATRX-positive cells were persistent in the mouse while the ATRX-positive tumor cells in the parental tumor were diffuse (Figure 3C,G). Hematoxylin & Eosin (H&E) staining in the PDX model indicated hallmarks of GBM with dense nuclei and pseudo-palisading necrosis which was present in the primary parental tumor from the patient (Figure 3D,H). This molecular characterization was conducted and consistent on 15/37 PDX models in Table 1 or the most malignant PDX sample, i.e., PDX samples that had a greater than 35% decrease in time to moribund condition from original to third passage. Additionally, in representative case 2409, the first two passages through a mouse were diffuse and infiltrative as indicated by both STEM 121 (Figure 4A,D) and human nuclear antigen staining (Figure 4B,E), respectively. This corresponded to the increased infiltration of the tumor growth area visualized with H&E staining (Figure 4C,F). After further implantation and passage, the cells became more localized into a uniform mass, similar to immortalized cell lines such as U87MG, as seen with STEM 121 and human nuclear antigen staining (Figure 4G,H), respectively. As human tumor cells were passaged through subsequent mice, some presented with a more “cell line” phenotype, with cells becoming more localized (Figure 3I) and less invasive. Further, the glial fibrillary acidic protein (GFAP) staining also indicated increased astrogliosis surrounding the tumor with each passage (Figure 5A–C). The last passage had similar GFAP astrogliosis to the tissue from the primary resection (patient parental tumor) which demonstrated moderate gliosis (Figure 5D). In this PDX line, the animals became moribund with the first intracranial implant at 78 days post implant, 72 days the second time, and 71 days the 3rd passage.

Alternatively, patient-derived line 2025 (see Figure 6) remained invasive and failed to develop a solid tumor mass with subsequent intracranial passages (Figure 6A,D,G). Tumor line 2025 was developed from a 58-year-old patient undergoing resection after biopsy confirmed GBM. STEM 121 cytoplasmic human staining indicated a more cortical development of tumor in the third passage (Figure 6D). However, the GFAP-reactive gliosis was very strong in all serial passages, similar to scar formation in a human GBM. The first intracranial injection (Figure 6G) indicated a global GFAP reaction throughout the brain very similar to the tumor morphology in the patient sample (Figure 6I). The third 2025 passage through a mouse had a more localized GFAP gliosis reaction responding to tumor engraftment (Figure 6H). Patient-derived line 2025 impaired mice 45% more quickly from the 1st passage to 3rd passage. The mice became moribund at 156 days, 120 days, and 85 days post implantation for passages 1, 2, and 3, respectively, demonstrating increasing malignancy with serial passage.

Visualization of cellular growth in the mouse intracranial model is difficult to track in live animals due to signal intensity and the imaging paradigms currently available. For instance, with small animal MRI it is difficult to image small intracranial and invasive tumors due to the sensitivity of the technique, which does not pick up the highly invasive and the often diffuse infiltrating cells which are a hallmark of GBM. Utilization of lentiviral genomic expression of fluorescent tags in addition to commercially-available cell tracker dyes to visualize tumor growth in FFPE sections gives inconclusive data, possibly due to the processing of the sample and signal strength of the markers. We determined immunohistochemistry to be the best option for tumor visualization and validation of human tumor cells within the mouse brain using human-specific antigen markers.

In general, the orthotopic stem-cell-derived PDX models maintained a remarkably similar morphology throughout the first, second, and third in vivo intracranial passages, retaining markers for human antigens and molecular hallmarks of the parent tumor (IDH1 status) along with invasiveness similar to that seen in GBM. Utilizing cells with low passage numbers, both in vitro and in vivo, is imperative to maintain the genomic aberrations, cellular heterogeneity, and molecular characteristics associated with GBM and avoid drift from the original sample. Our study is the first to recapitulate a parental tumor in a mouse model and descriptively compare the parental tumor to the serial passages in vitro and in vivo. Furthermore, passage through the brain versus flank is important to study the unique microenvironment associated with primary GBM malignancy and progression. GFAP, a marker of astrocytic cells, is evident in both the primary and corresponding PDX intracranial xenografts, verifying a similar behavior of the cells within the xenograft (data not shown). The PDX-model-activated astrocytes are characteristically present surrounding and infiltrating the tumor indicating gliosis, a localized response to damage caused by the tumor. These astrocytes are predominately derived from the mouse as significant overlap with human cytoplasmic and human nuclear antigen markers is not prominent, although this will need to be investigated further. Overall, the invasive nature of the low passage PDX intracranial models are characteristic of the primary human GBM, with the surrounding microenvironment contributing to the malignancy of the tumor. Further studies will, by utilizing intracranial PDX models, answer many questions as to what cell types contribute to tumor growth and maintain the invasive nature of the tumor.

## 3. Materials and Methods

### 3.1. Tumor Collection

Tumor tissue was collected directly from the operating room according to approved ethics guidelines and informed consent to the University Hospitals Medical Center Cleveland, OH, USA. A portion of the tissue was formalin-fixed and paraffin embedded (FFPE) for immunostaining and histological analysis. The remaining tumor tissue was dissociated into a single-cell suspension with a human tumor dissociation kit (Miltenyi Biotec Inc., 130-095-929, Auburn, CA, USA). Briefly, tissue was chopped into 2–4 mm pieces and transferred to an RBC lysis media (Lonza) for 5 min at room temperature in a tissue culture dish. The tissue was then washed with RPMI 1640 media (Gibco Invitrogen 11875-093, Waltham, MA, USA). The tissue was then transferred to gentleMACS C tubes (Miltenyi Biotec Inc., 130-093-237, Auburn, CA, USA) containing RPMI 1640 media and the enzymatic mixture from the tumor dissociation kit. Tissue was dissociated on the gentleMACS dissociator (Miltenyi Biotec, Inc., 130-093-235, Auburn, CA, USA) using the “human tumor” program. Single cells were then filtered through a 70 μm cell strainer (Fisher Scientific, 22363548, Hampton, NH, USA) and pelleted. Cells were maintained in culture for less than a week in serum-free Neural stem cell (NSC) complete medium (see below) prior to engraftment intracranially. Cells at this point were in single cells and clumps resembling “loose neuropheres”.

Cells were kept in a 5% CO_2_, 37° humidified incubator in MACs neuro medium with Neurobrew-21 (Miltenyi Biotec Inc., 130-093-570 and 130-097-263, respectively, Auburn, CA, USA), 20 ng/mL epidermal growth factor (EGF) (Peprotech, AF-100-15, Rocky Hill, NJ, USA) and 20 ng/mL fibroblast growth factor (FGF) (Peprotech, 100-18B, Rocky Hill, NJ, USA) with penicillin–streptomycin (Gibco Invitrogen, 15140122, Waltham, MA, USA) and l-glutamine (Gibco Invitrogen, 25030081, Waltham, MA, USA) (serum-free NSC complete media). Cells were grown in a suspension flask (CytoOne, CC-672-4175, USA Scientific, Oscala, FL, USA) and complete media was added every third day. Cultured cells were pelleted and re-suspended to 2 × 10^5^ cells per 3 μL in growth medium and placed on ice prior to implantation.

### 3.2. Xenografting

Patient-derived glioma stem-like cells were implanted intracranially into NOD scid gamma (NSG) mice to establish the PDX models. Immunosuppressed animals (NOD scid gamma, Jackson Labs, stock #005557, NSG, Bar Harbor, MA, USA), 4–6 weeks of age, males and females, were utilized to avoid rejection of the implanted cells. For each patient sample 2–3 mice were implanted intracranially. Briefly, animals were placed under anesthesia (inhaled isofluorane). Once fully anesthetized, lidocaine was applied and a small incision was made through the scalp and the bregma was identified. A small 25-gauge burr hole was made 2 mm caudal and 3 mm to the right of bregma. A 22-gauge Hamilton syringe (Fisher Scientific, 88011, Hampton, NH, USA) was inserted and placed 3 mm below the skull and then retracted 0.5 mm to establish a pocket for implantation of cells using a stereotactic frame. Cells, 2 × 10^5^ cells per 3 μL, were then slowly injected into the area and the Hamilton syringe was held in place for 3 min post injection to prevent reflux. The hole was sealed with bone wax and the incision was closed with surgical glue and non-dissolvable sutures. Animals were given analgesia and maintained on a heating pad until recovery.

Care and housing of the animals was provided by the University Animal Resource Center following Institutional Animal Care and Use Committee (IACUC) oversight. The facility follows recommendations from the Guide for the Care and Use of Laboratory Animals of the National Institutes of Health. Mice were kept in a specific pathogen-free environment in microisolator cages (two animals per cage) and exposed to 12-h light/12-h darkness cycles with standard food and water *ad libitum*. Mice were weighed weekly and checked daily for tumor growth symptoms according to the IACUC tumor burden policy.

For establishment of patient-derived xenografts (PDX) tumor models, animals were sacrificed at signs of moribund (i.e., declining mobility and grooming along with weight loss), and the whole brain harvested for tumor cell isolation or FFPE. To derive the next generation of GSCs, mouse cells were removed from the culture utilizing the mouse cell depletion kit from Miltenyi Biotec. Briefly, tissue was dissociated using 0.25% trypsin (Gibco Invitrogen, 25200-056, Waltham, MA, USA) via the gentleMACS dissociator (Miltenyi Biotec, Inc.,130-093-235, Auburn, CA, USA) using the implanted “mouse tumor” program. Following dissociation and filtration into single cells, cells were maintained in culture on suspension plates (CytoOne, CC7672-4175, Oscala, FL, USA) in serum-free complete media for less than two weeks prior to re-implanting. An aliquot of cells was either frozen for later analysis or spun down and counted to 5 × 10^4^ to 2 × 10^5^ cells per 3 μL of serum-free NSC. Complete media were again implanted following the procedure outlined previously. These human tumor cells were then engrafted again into at least two NGS mice indicated by the “ic” passage number (ic1, ic2, or ic3) following the procedure outlined previously.

### 3.3. Immunohistochemistry and H&E Staining

Primary GBM or PDX samples were fixed in 10% buffered formalin, embedded in paraffin, sectioned at 5 μm, and mounted on Superfrost^®^ Plus slides (Fisher Scientific, 12-550-15, Hampton, NH, USA). Sections were then hydrated through descending ethanol to water. Endogenous peroxidase activity was eliminated by incubation in 3% H_2_O_2_ for thirty minutes prior to heat-induced epitope retrieval (HIER). HIER was performed using a citrate-based retrieval buffer, pH 6.1 (Dako, S1699, Santa Clara, CA, USA) for 10 min in a 96 °C water bath. The mouse antigen blocking kit (Vector laboratories, PK-2200, Burlingame, CA, USA) was utilized to reduce background staining according to manufacturer protocols. To reduce non-specific binding, sections were incubated in 10% normal goat serum (Fisher Scientific, PCN5000) in Tris-buffered saline, (TBS; 50 mM Tris-HCl 150 mM NaCl, pH 7.6, Bio-Rad, 170-6435, Hercules, CA, USA) for 30 min prior to application of the primary antibody. Antibodies used in this study were mouse monoclonal antibody specific for human cytoplasmic marker, STEM 121 (Takara Bio, Y40410, Mountain View, CA, USA); mouse monoclonal antibody to human nucleoli (Abcam, ab190710, Cambridge, UK); mouse monoclonal antibody to human glial fibrillary acidic protein (GFAP, Abcam, ab10062, Cambridge, UK; reacts with mouse, rat, and human); rabbit polyclonal alpha-thalassemia/mental retardation, X-linked antibody (ATRX, Sigma-Aldrich, HPA001906, St. Louis, MO, USA); mouse monoclonal antibody to isocitrate dehydrogenase 1 R123H mutation (IDH1R132H, Dianova, DIA-H09, Warburgstrabe, Hamburg, Germany); and rabbit monoclonal Ki67 antibody (Cell Signaling Technologies, 9027S, Danvers, MA, USA). Immunohistochemistry was visualized via the peroxidase–anti-peroxidase method using 3.3’-diaminobenzidine (DAB) as a chromogen (Thermo Scientific, TA-125-QHDX, Waltham, MA, USA) [19]. Serial sections were stained with hematoxylin and eosin to highlight the areas of tumor cell growth.

## 4. Conclusions

Our collection of PDX intracranial models will allow for a better understanding of the complexity of GBM and its development by testing the genomics of the models and assessing therapies. Ongoing and future studies will establish PDX models in humanized animals in order to get a better picture of cellular contribution to GBM invasiveness and malignancy, aiding in the study of immune modulators and cellular therapies as well as other potential therapies. Individual patient PDX models could eventually be used to model responses to various treatment modalities. Thus, our intracranial PDX model is a valuable tool to promote the advancement of GBM research by allowing an environment that closely mimics that of GBM.

## Figures and Tables

**Figure 1 ijms-21-05113-f001:**
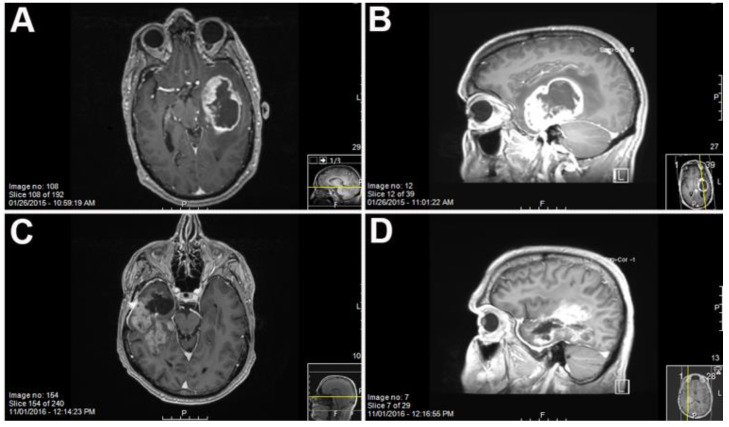
MRI of patient 2025, a 58 year old male ((**A**) coronal and (**B**) sagittal) and patient 2409, a 57 year old male ((**C**) coronal and (**D**) sagittal) with T1 weighted gadolinium contrast showing the location of the Glioblastoma multiforme (GBM) prior to resection.

**Figure 2 ijms-21-05113-f002:**
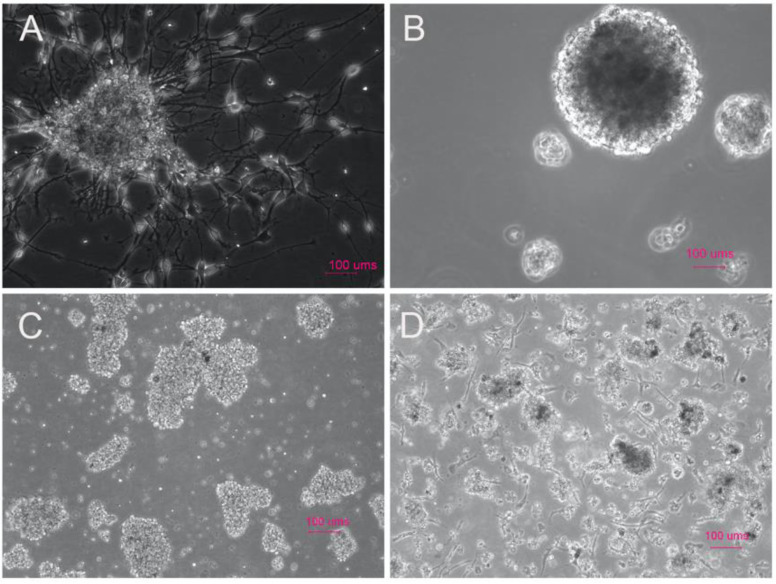
Patient-derived glioma stem cells (GSCs) from histologically confirmed GBM less than 7 days post dissociation in serum-free media demonstrating a wide range of phenotypes which all resulted in intracranial tumor growth in NOD scid gamma (NSG) mice. (**A**) Patient-derived cells from case 2025, a newly-diagnosed GBM; (**B**) patient-derived cells from case 2033, a multifocal GBM; (**C**) patient-derived cells from case 2188, a recurrent GBM; and (**D**) patient-derived cells from case 2216 from a newly-diagnosed GBM. All four have distinguishing characteristics of adherent neurospheres (**A**), large 3-dimensional neurospheres (B), and small clumps of cells with adherent cell populations (C,D) However, all form orthotopic xenografts which grow at increasing rates in successive generations. (Scale bar = 100 μm).

**Figure 3 ijms-21-05113-f003:**
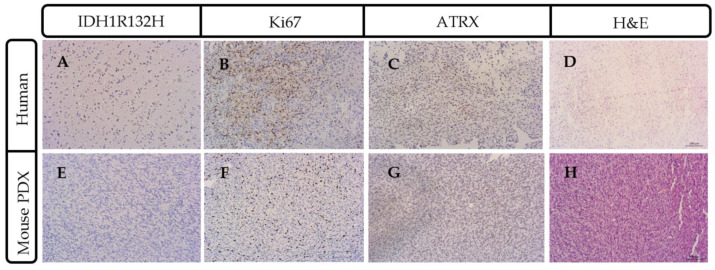
Parental patient tumor mutational status was recapitulated in PDX mouse model. Representative case 2409 lacked IDH1R132H consistent with the wildtype IDH1 molecular characterization (**A**) that was also present in the mouse model of glioma (**E**). Ki67 staining indicated a highly-proliferative tumor in the parental (**B**) and mouse PDX (**F**) model. ATRX (ATP-dependent helicase, X-linked protein) immunohistochemistry had a similar pattern of expression in the human (**C**) and the mouse (**G**). (**D**) ATRX and hematoxylin and eosin (H&E) staining of parental human patient tumor and (**H**) dense nuclei with pseudo-palisading necrosis is present in the mouse intracranial PDX model. Scale bar = 100 μm.

**Figure 4 ijms-21-05113-f004:**
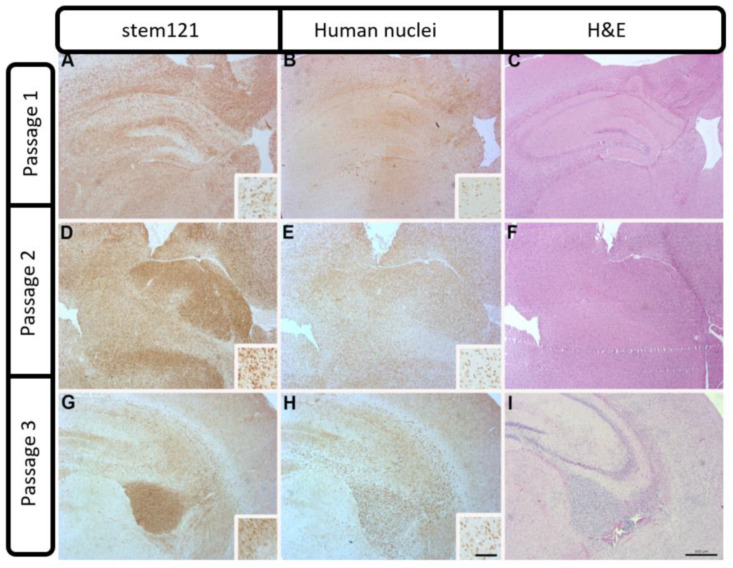
Patient-derived PDX model of case 2409 showing diffuse STEM 121 staining of human-derived cells in passages 1 and 2 (**A**,**D**), and a more localized staining with some invasion in passage 3 (**G**). An adjacent section of human nuclei staining showed the same pattern of diffuse staining in passages 1 and 2 (**B**,**E**) and a localized staining with some invading cells evident in passage 3 (**H**). Increased cellularity and infiltration of proliferating tumor cells were apparent in H&E stained sections Insets (**C**,**F**,**I**) show individual cellular staining. Scale bar = 500 μm. Inset scale bar = 100 μm).

**Figure 5 ijms-21-05113-f005:**
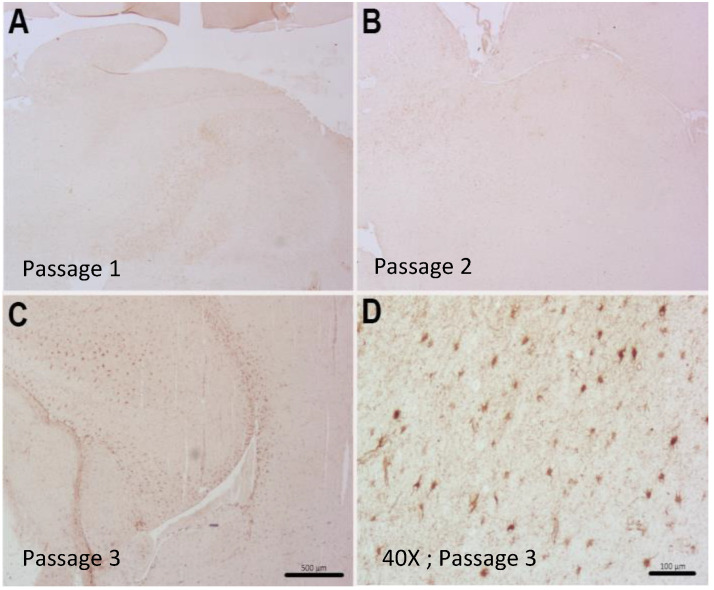
Patient-derived intracranial PDX model of case 2409 showed reactive astrogliosis with glial fibrillary acidic protein (GFAP) staining in the third passage (**C**) and little to no glial staining in the first two passages (**A**,**B**). (**D**) Tumor sample from the patient demonstrating reactive gliosis. Scale bars = 500 μm (**A**–**C**) and 100 μm (**D**).

**Figure 6 ijms-21-05113-f006:**
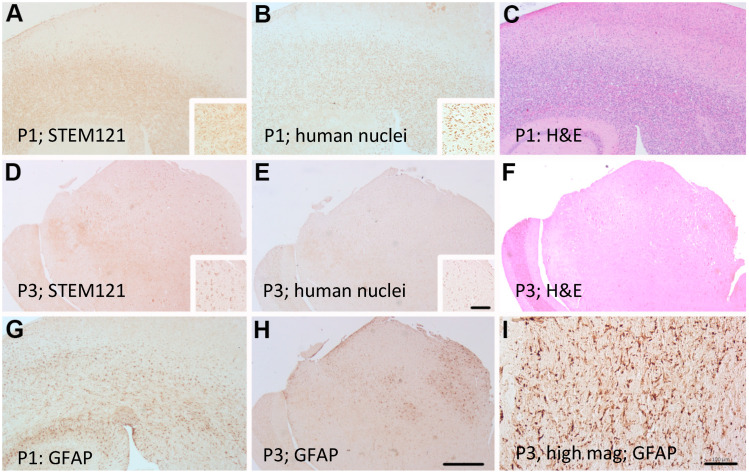
Adjacent sections of patient derived PDX model of case 2025 showing diffuse STEM 121 staining in passages 1 (P1) and 3 (P3) (**A**,**D**) respectively and diffuse human nuclei staining in passage 1 (P1) and 3 (P3) (**B**,**E**) respectively. GFAP staining showed reactive astrogliosis (**G**,**H**). Resected human tumor sample from the patient illustrates reactive gliosis (I). H&E staining clearly shows the increase tumor cell proliferation (**C**,**F**). Insets show individual cellular staining. Scale bar = 500 μm (**A**–**H**). Scale bar = 100 μm (**I**) Inset scale bar = 100 μm.

**Table 1 ijms-21-05113-t001:** Characteristics of GBM patient-derived intracranial xenografts. The table lists the 37 patient-derived xenograft (PDX) models which contained enough viable cells for at least three intracranial passages and represents about 54% of the patient samples excised. Each column is an average growth rate in number of days from the initial implant of the cells until moribund (*n* = 2). The last column shows the percent decrease in time for symptom development from the first to the last passage. Eleven cases demonstrated a 40% or more decrease in days as the passage number increased. Dashes represent cases where no increase in symptom development was found in subsequent tumor cell passage. No correlation between age at diagnosis or gender was determined to be a factor in the take rate of the cells.

Case	Patient Age at Diagnosis, Gender	Molecular Status of Patient Tumor (If Known)	Survival: Original PDX (in Days Post Inoculation)	Survival 2nd Generation PDX (in Days Post Inoculation)	Survival 3rd Generation PDX (in Days Post Inoculation)	Percent Decrease in Time to Moribund (from Original to 3rd Passage)
1711	75, male	IDH1wt	163	107	118	28%
1768	51, male	IDH1wt	131	130	102	22%
1783	75, male	N/A	138	93	72	48%
1786	67, male	IDH1wt	93	60	53	43%
1892	63, female	N/A	126	143	109	13%
1914		N/A	112	158	100	11%
1918	63, female	IDH1wt	90	152	50	44%
1919	53, male	IDH1wt, ATRXwt	85	119	135	-
1949	N/A	IDH1wt	149	122	109	27%
1951	N/A	N/A	148	86	102	31%
1953	N/A	IDH1wt	166	142	109	34%
1959	N/A	N/A	94	49	34	64%
1962	N/A	IDH1wt	154	51	86	44%
1963	N/A	N/A	154	89	109	30%
1997	N/A	N/A	90	107	96	-
2014	N/A	N/A	88	115	112	-
2025	N/A	N/A	156	120	85	45%
2033	N/A	N/A	151	131	106	30%
2070	N/A	IDH1wt	148	125	92	38%
2072	N/A	IDH1wt	167	113	63	62%
2078	N/A	IDH1wt	161	113	108	33%
2091	N/A	IDH1wt	166	110	95	43%
2095	N/A	N/A	154	79	89	42%
2096	N/A	N/A	135	16	106	22%
2114	N/A	IDH1wt	112	110	100	-
2142	36, female	IDH1mut, ATRXmut	154	74	95	38%
2144	N/A	IDH1wt, ATRXwt	139	113	92	34%
2152	79, male	IDH1wt, ATRXwt	152	118	82	46%
2188	N/A	IDH1wt	115	90	92	20%
2187	35, male	N/A	65	97	82	-
2214	N/A	IDH1wt, ATRXwt	83	97	82	-
2216	71, male	IDH1wt, ATRXwt	155	102	36	77%
2273	59, female	IDH1wt, ATRXwt	42	93	78	-
2284	64, female	IDH1wt, ATRXwt	40	17	99	-
2302	60, male	IDH1wt	127	71	85	33%
2381	67, male	IDH1wt, ATRXwt	92	101	71	23%
2409	57, male	IDH1wt, ATRXwt	78	72	71	9%

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
