# Peer review of "Generation of Glioblastoma Patient-Derived Intracranial Xenografts for Preclinical Studies"

_ijms, 2020, doi:10.3390/ijms21145113_

Round 1
Reviewer 1 Report
In the manuscript "Generation of Glioblastoma Patient Derived Intracranial Xenografts for Preclinical Studies", authors have reported the development of the PDX model of glioblastoma and suggested it as a novel tool for personalized medicine. The idea behind this is great and can be really helpful if issues related to the model can be fixed.
I have some concerns regarding their study:
- There is some published manuscript that reported the development of the intracranial PDX mouse model of glioblastoma, authors should show the novelty of their study over them: PMID: 30992429 and PMID: 29321663.
- In the above-mentioned manuscript, Lee et al. reported their observation that takes about 4 months to be developed that make this model hard to be used for personalized medicine as glioblastoma patient have low survival rate about 15 months. Also, for developing the model researchers have to use immunodeficient mice that make the microenvironment of the tumors different from the brain. To achieve their conclusion, the authors need to have a convincing rationale.
- I suggest the authors add a figure to show the developed tumors in the mice before or after extraction to show the successfully of procedure.
- I suggest the authors add more information about the timeline of the whole process.
Author Response
MDPI review
Reviewer 1.
Point 1.
There is some published manuscript that reported the development of the intracranial PDX mouse model of glioblastoma, authors should show the novelty of their study over them: PMID30992429 and PMID:29321663
Response:
Thank you for your suggestion. The Lee et al. manuscript investigated glioma models in the eye, which allows for ease of treatment but does not take into account the microenvironment of the brain. In the Loskutov et al. paper, they investigated LPA signaling in primary cilium in GBM. They utilized a similar GBM model but with high passage number. Our study investigates the development of low passage models for the interrogation of new therapies, which will have more relevance to the treatment of patients. Further, we have shown common GBM mutations are preserved from the patient GBM tumors to the PDX mouse model.
Point 2.
In the above mentioned manuscript, Lee et al. reported their observation that takes about 4 months to be developed that makes this model hard to be used for personalized medicine as glioblastoma patient have low survival rate of 15 months.
Response:
While the development of the PDX model does indeed take at least 90 days, this would allow for treatment strategies to be investigated in similar samples or with a recurrent sample. In some patients, diagnosed at less than 55 years of age, whom will likely recur, this will be essential in determining an effective treatment strategy. The time of our models becoming moribund or deceased in represented in Table 1.
Point 3.
Also, for developing the model researchers have to use immunodeficient mice that make the microenvironment of the tumors different from the brain. To achieve their conclusion, the authors need to have a convincing rationale.
Response:
This is certainly a valid point and caveat in the PDX models developed, however, we think this is a great start to the development of effective treatment strategies by utilizing low passage glioma lines instead of the less clinically relevant high passage immortalized cell lines. In order to study the role of immunotherapies a different model must be developed, such as humanized mice, however, that is currently a costly venture.
Point 4.
I suggest the authors add a figure to show the developed tumors in the mice before or after extraction to show the successfully of procedure.
Response:
We had added H&E of the parental tumor and the developed PDX tumor. This shows many hallmarks of GBM such as dense tumor nuclei, necrosis and pseudopalisading necrosis.
Point 5.
I suggest the authors add more information about the timeline of the whole process.
Response:
Thank you for your suggestion. We had added more details in Table 1 as to the timeline for the process of the development of our PDX lines.
Reviewer 2 Report
In this manuscript the authors present a well-written, thoroughly documented description of their experience generating orthotopic PDX models from patients with glioblastoma at one institution. Their methods are clear and these 37 orthotopic PDX models have the potential to be a valuable resource to the neuro-oncology community. I fully agree with the introduction, assessment of the limitations of current in vitro and in vivo models, and approach to establishing the xenografts. Unfortunately, the manuscript falls short of fully characterizing these models, limiting their utility in its current form.
The characterization of these xenografts is based on tumor growth as defined by time in the mouse. There is no description of the parental genomic or expression pattern, the growth of the parental tumor in a human, nor whether the xenograft recapitulates the parental tumor on a histologic, genomic, or expression level. A subset of these data would be very helpful for using these models in the future. One way to incorporate some information that already exists would be to include a table with some descriptive data from each patient (age at diagnosis, overall survival, whether the tumor was obtained at diagnosis or progression, overall survival, etc) along with any molecular characteristics that are known, likely from the IHC (as some of these tumors were likely obtained before routine molecular testing was available and clinical tissue may not be obtainable for this study).
The authors describe the fact that the tumors express 2 different growth patterns. It would be very interesting to know which xenografts (and what proportion) exhibit which pattern and how it changed over time. It’s unclear from the manuscript what proportion of the xenografts were phenotyped.
It would be helpful to know whether and how well the xenografts recapitulate the tissue structure of the parent tumor. The authors mention comparison to the human tumor in some specific xenografts, but it might be helpful if these images were shown for comparison. Moreover, immunostaining for Ki67 or another marker of proliferation could be helpful in supporting the authors’ claim of growth acceleration through serial passages in the mouse.
Minor points:
Figure 4 - I'm a little unclear on how to interpret this figure. Is this GFAP human-specific or mouse specific? Are the authors showing human-derived astrocytes remain in the xenograft and become reactive, or that the mouse is developing gliosis in reaction to the xenograft?
In some of the figures, it might be helpful improve readability to label the stains on the figure X & Y axes (Passage #, Stem121, GFAP, H&E, etc).
Figure 5 – Please clarify whether panels 'G' and 'H' are from passage 1 or 3?.
Please clarify which Stem121 antibody you used. The antibody number in the manuscript is not a CST antibody
Author Response
MDPI- reviewer 2
Point 1.
There is no description of the parental genomic or expression pattern, the growth of the parental tumor in a human, nor whether the xenograft recapitulates the parental tumor on a histologic, genomic, or expression level. A subset of these data would be very helpful for using these models in the future. One way to incorporate some information would be include a table with some descriptive date from each patient along with molecular characteristics that are known.
Response:
Thank you for your great suggestion. We have added details to Table 1. We have also characterized 15/37 PDX lines which had a greater than 35% decrease in survival from passage 1 to 3 for common glioma mutations. We added a new Figure 3 that shows representative images from case 2409 with IDHR132H and ATRX mutational status, in addition to Ki67 and H&E from human and mouse PDX samples.
Point 2.
The authors describe the fact that the tumors express 2 different growth patterns. It would be interesting to know which xenografts and which portion exhibit which pattern and how it changed over time. It is unclear from the manuscript what portion of the xenografts were phenotyped.
Response:
We have noticed a majority of the PDX samples grow as diffuse tumors (80%) and it is rare to have a low passage line to grow as bulk tumor (such as U87MG). For bulk tumor growth, it would typically need to be passaged >3 times. We had phenotyped all 37 samples for this assessment and have added these details to the manuscript.
Point 3.
It would be helpful to know whether and how well the xenografts recapitulate the tissue structure of the parent tumor. The authors mention comparison to the human tumor in some specific xenografts, but it might be helpful if these images were shown for comparison. Moreover, immunostaining for Ki67 or another marker of proliferation could be helpful in supporting the authors’ claim of growth acceleration through serial passages in the mouse.
Response:
We have incorporated your suggestion into a new Figure 3. We were able to show areas of increased proliferation in the human parental tumor which was recapitulated in the mouse PDX line.
Minor points:
Point 1.
Figure 4. I’m a little unclear on how to interpret this figure. Is this GFAP human-specific or mouse specific? Are the authors showing human derived astrocytes remain in the xenograft and become reactive, or that the mouse is developing gliosis in reaction to the xenograft?
Response:
We had similar thoughts. This particular antibody recognizes human, mouse and rat GFAP (mouse monoclonal antibody to human glial fibrillary acidic protein (GFAP, Abcam, ab10062, Cambridge, UK). Therefore, it may be a combination of human derived astrocytes or the mouse developing gliosis. This conclusion would have to be investigated further but was not part of our study.
Point 2.
In some of the figures, it might be helpful to improve readability to label the stains on the figure X and Y axes (Passage #, stem121, GFAP, H&E, etc).
Response:
Thank you for the suggestion. I have added descriptors to the figure to make this easier to read.
Point 3.
Figure 5- please clarify whether panels G and H are from passage 1 or 3?
Response:
Thank you for pointing that out. I have added descriptors to each panel of this figure.
Point 4.
Please clarify with Stem121 antibody you used. The antibody number in the manuscript is not a CST antibody.
Response:
We apologize for the error. We have added the correction for the STEM 121 antibody that was used in this manuscript, which was human cytoplasmic marker, STEM 121 (Takara Bio, Y40410, Mountain View, CA).
Reviewer 3 Report
Kerstetter-Fogle et al establish orthotopic patient-derived xenografts from GBMs that can be propagated in vivo while resembling the primary tumor. The study is important as clinically relevant intracranial PDX models are currently lacking in the glioma field. Overall this is an interesting and laborious study that generated a cohort of glioma orthotopic PDX models. I think several points could be included to further increase the impact of the work:
- Pathology of the tumors (Grade, primary/recurrence, relevant mutations/CNVs) would need to be included. Perhaps a statistical analysis to determine whether there is a clinical or genetic aspect of tumors that are more likely to 'take' in vitro or in vivo
- Further stainings (IHC, FISH, etc) comparing the primary tumor and the in vivo propagated tumor with relevant markers. For example if the primary tumor is EGFR amplified, is this reflected in the PDX model?
- How molecularly similar are the in vitro models and the derivative in vivo orthotopic PDX models - taking a subset of the generated models and checking whether original oncogenic alterations are maintained and/or performing RNA-seq of paired samples (if this is feasible with the current lockdowns).
Author Response
MDPI- Reviewer 3 reply
Point 1.
Pathology of the tumors (Grade, primary/recurrence, relevant mutations/CNVs) would need to be included. Perhaps a statistical analysis to determine whether there is a clinical or genetic aspect of tumors that are more likely to “take” in vitro or in vivo.
Response:
Thank you for your suggestion. We have incorporated the relevant mutations and all the tumors isolated were Grade IV glioma or GBM. This has been also incorporated into the text.
Point 2.
Further stainings (IHC, FISH, etc) comparing the primary tumor and the in vivo propogated tumor with relevant markers. For example, if the primary tumor is EGFR amplified, is this reflected in the PDX model?
Response:
Thank you for your insightful suggestion. We have added a new figure (Figure 3) that is a representative sample of 15/37 matched human/mouse samples that had a greater than 35% decrease in survival from passage 1 to 3 for IHC characterization. This is IHC of IDH1R132H, ATRX, Ki67 and the H&E staining of the 2409 human and mouse sample. These markers are common markers used in the clinic for prognosis.
Point 3.
How molecularly similar are the in vitro models and the derivative in vivo orthotopic PDX models?
Response:
We had added a new figure which investigates the common mutations in GBM and whether this had translated to our PDX model. We chose the 2409 PDX line but this was appropriately similar across the 15/37 samples investigated.
Point 4.
Taking a subset of the generated models and checking whether original oncogenic alterations are maintained and/or performing RNA-seq of paired samples (if this is feasible with the current lockdowns).
Response:
We were unable to perform RNA-Seq on these samples due to the current pandemic and time line. However, we will likely investigate this in the future. Thank you for the suggestion.
Round 2
Reviewer 1 Report
I appreciate the time authors spend on answering the comments.
I still have some concerns regarding their response to my comments.
- When I asked about the advantages of their model over the two already published papers in this file, so they should talk about it in the discussion part of the manuscript besides mentioning them in the introduction to show they have done a complete review of the literature.
- To show the success of their procedure they should show an image of developed tumours (prior to extraction using fluorescent imaging in the mice or after extraction using a simple photo, it should show the size of tumors after the specific time of implantation of the cells.
Author Response
Point 1. When I asked about the advantages of their model over the two already published papers in this file, so they should talk about it in the discussion part of the manuscript besides mentioning them in the introduction to show they have done a complete review of the literature.
Response: Please see discussion section with this amendment added.
Point 2. To show the success of their procedure they should show an image of developed tumours (prior to extraction using fluorescent imaging in the mice or after extraction using a simple photo, it should show the size of tumors after the specific time of implantation of the cells.
Response: Please see paragraph beginning on line 237 on the difficulty of imaging diffuse growth of the primary tumors. I have uploaded a MRI image of primary tumor 1959 demonstrating the difficulty with imaging (please see attachment). We are trying to develop a reporter system in order to capture the growth pattern of the tumors.

Reviewer 3 Report
The authors have addressed all my comments.
Author Response
Thank you for your help with this manuscript.